# Comparison of the Determination of Some Antihypertensive Drugs in Clinical Human Plasma Samples by Solvent Front Position Extraction and Precipitation Modes

**DOI:** 10.3390/molecules28052213

**Published:** 2023-02-27

**Authors:** Kamila Jaglińska, Beata Polak, Anna Klimek-Turek, Emilia Fornal, Anna Stachniuk, Alicja Trzpil, Robert Błaszczyk, Andrzej Wysokiński

**Affiliations:** 1Department of Physical Chemistry, Medical University of Lublin, Chodźki 4A, 20-093 Lublin, Poland; 2Department of Bioanalytics, Medical University of Lublin, Jaczewskiego 8b, 20-090 Lublin, Poland; 3Chair and Department of Cardiology, Medical University of Lublin, Jaczewskiego 8, 20-090 Lublin, Poland

**Keywords:** clinical sample preparation, solvent front position extraction, precipitation, LC-MS/MS analysis, antihypertensive drugs

## Abstract

The determination of the selected antihypertensive drugs in human plasma samples with the novel solvent front position extraction (SFPE) technique is presented. The SFPE procedure combined with LC-MS/MS analysis was used for the first time to prepare a clinical sample containing the drugs mentioned above from different therapeutic groups. The effectiveness of our approach was compared with the precipitation method. The latter technique is usually used to prepare biological samples in routine laboratories. During the experiments, the substances of interest and the internal standard were separated from other matrix components using a prototype horizontal chamber for thin-layer chromatography/high-performance thin-layer chromatography (TLC/HPTLC) with a moving pipette powered by a 3D mechanism, which distributed the solvent on the adsorbent layer. Detection of the six antihypertensive drugs was performed by liquid chromatography coupled to tandem mass spectrometry (LC–MS/MS) in multiple reaction monitoring (MRM) mode. Results obtained by SFPE were very satisfactory (linearity R^2^ ≥ 0.981; %RSD ≤ 6%; LOD and LOQ were in the range of 0.06–9.78 ng/mL and 0.17–29.64 ng/mL, respectively). The recovery was in the range of 79.88–120.36%. Intra-day and inter-day precision had a percentage coefficient of variation (CV) in the range of 1.10–9.74%. The procedure is simple and highly effective. It includes the automation of TLC chromatogram development, which significantly reduced the number of manual operations performed, the time of sample preparation and solvent consumption.

## 1. Introduction

Resistant hypertension is a persistent blood pressure value higher than 140/90 mm Hg. Such a clinical situation occurs when proper blood pressure control cannot be achieved, despite the appropriate treatment, which uses three antihypertensive drugs, adequately combined and in total doses. Moreover, high pressure is confirmed in 24-h blood pressure monitoring. Additionally, non-compliance with medical recommendations is excluded [1,2]. The most reliable method of assessing compliance in patients suffering with resistant hypertension is monitoring the concentration of antihypertensive drugs in body fluids [3,4]. The data from relevant literature indicate that patients classified as those with the so-called resistant arterial hypertension, in more than 80% of cases, do not take at least one antihypertensive drug. Moreover, in about 13% of such patients, it was impossible to detect the presence of an antihypertensive drug in the blood serum [5,6]. Thus, the presence, absence, or any change in concentration of some substances measured in the blood may help identify patients suffering from so-called pseudo-hypertension. Such a mode can also avoid costly diagnostics for secondary forms of hypertension [7,8].

To monitor the concentration of antihypertensive drugs in the patient’s blood, high-performance liquid chromatography coupled with tandem mass spectrometry (LC-MS/MS) is usually used [9,10,11]. Sample preparation is the most crucial step in quantitative analysis. The correctly performed stage guarantees the repeatability and reliability of the results and allows the avoidance of problems related to contamination of the measuring device [12]. In the case of biological samples of antihypertensive drugs, the most often used preparation method is precipitation [13,14,15]. It is a practical, easy-to-make, and low-cost method, but is not free from disadvantages. A relatively poor purification of samples is one of them [16]. Additionally, the matrix components of the sample may interfere with the correct course of the analysis (matrix effect) [12] and, in this manner, affect the result of the determination. Hence, it is essential to find a method of preparing biological samples that will primarily clean them from interfering substances without extending the time and cost of the analysis.

Our previous article proposed a modern approach to preparing biological samples of antihypertensive drugs [17]. This procedure is called solvent front position extraction (SFPE) [18] and can be successfully considered an exciting and effective alternative to the classical precipitation method. It is based on thin-layer chromatography/high-performance thin-layer chromatography (TLC/HPTLC) [18] and takes place in several stages. The internal standard (IS) is added to the sample solution in the first stage. Then, in the second stage, a drop of its solution is applied directly onto the start line of a chromatographic plate. The third step is the elution of substances, including the IS from the serum/plasma/blood zone, carried out over a short distance. In the fourth step, the chromatogram is developed in such a way as to obtain the R_f_ values for the substance/s and an IS of about 0.5. As a result, the target substances are separated from the matrix components, which possess higher and lower adsorption energy than the substances of interest. In the fifth stage, the chromatogram is developed to locate the substance/s of interest and IS zones at the final solvent front position. Finally, in the last step (sixth), the substance/s and IS are extracted from this area and transferred further to the quantification with instrumental techniques [18,19]. The investigated compounds concentrated in the final solvent front position are usually visible without derivatisation under a UV lamp so that they can be easily extracted from the adsorbent surface for further LC-MS/MS analysis. In 2021, the SFPE successfully separated six antihypertensive drugs (lercanidipine, ramipril, hydrochlorothiazide, valsartan, nebivolol and perindopril) from matrix components and quantitatively determined them [17]. The SFPE procedure was carried out using a prototype device equipped with a mobile pipette delivering solvent to the adsorbent layer, which allowed for its partial automation [17]. The previous article has shown that our procedure for preparing biological samples of antihypertensive drugs is accurate and repeatable, and the standard deviation (%RSD) did not exceed 6% [17]. 

Additionally, low solvent consumption proves its social impact on the environment and its low research costs [17]. Other advantages of SFPE are the fast sample preparation time, as well as the high selectivity of the method. However, real samples taken directly from the patient containing the antihypertensive drug still need to be investigated. Hence, in this report, we used the SFPE procedure to prepare the plasma samples mentioned above before their quantification. Such a step is necessary when the method is allowed to prepare biological samples.

This article will discuss the new mode of sample drug preparation, SFPE, for real samples from clinical institutions (medical wards). Moreover, the application of SFPE to determine antihypertensive drugs (the short description of the substances studied is provided in Section 3.1) in real serum samples will be presented for the first time. The innovative approach and the parameters affecting the SFPE performance will also be explained. In addition, the unique characteristics that make SFPE a credible alternative to the precipitation mode in this area will be discussed. Moreover, the application of the gradient HPLC elution combined with the SFPE for the quantification of an antihypertensive drug was presented for the first time.

## 2. Results

### 2.1. Precipitation Procedure of Sample Preparation

Precipitation is currently the most frequently used method for the preparation of antihypertensive drugs originating from biological samples [20,21,22,23,24,25]. It is not very complicated or expensive compared to SPE (solid phase extraction), where selecting appropriate columns is a big challenge. Additionally, the latter procedure is costly and complex [26,27,28]. Therefore, we chose precipitation as a reference method for our proposed SFPE procedure. Comparing the quantitative results, time, and costs incurred will evaluate the suitability of the proposed approach in preparing complex clinical samples. This step is necessary if the method is to be used in routine testing.

The first step before quantifying selected antihypertensive drugs in clinical samples was the preparation of calibration curves on the reference plasma (details in Section 3.4). LC-MS/MS made it possible to obtain peak signals on this basis of a relationship of the relative concentration of the ratio of analyte/IS prepared as a function of the ratio of the peak area of the analyte and the IS. A linear relationship was obtained for each substance, described by the equation (Table 1).

The calibration curves developed for antihypertensive drugs have a coefficient of determination (R^2^) value in the range of 0.997 to 0.999. This proves a reasonable adjustment of the curve to empirical points, thus confirming the effectiveness of this method.

The next step included testing real samples of three mixtures with selected medications. The division of drugs into groups was made on the basis of their currently used combinations in the treatment of cardiovascular problems [29,30]. Finally, drug concentrations (ng/mL) in the clinical sample were determined by measuring the analyte/IS peak area ratio. The results are presented in Table 2.

The values of the tested drugs’ relative standard deviation (%RSD) oscillate around 5%, proving the method’s precision. The limit of quantitation (LOQ) and the limit of detection (LOD) was calculated using the formulae: LOD = 3.3σ/s and LOQ = 10σ/s, respectively, where σ is the standard deviation of the response, and s is the regression line slope. The values of the limit of detection and quantification of the determined compounds were in the range of 0.02–4.41 ng/mL and 0.05–13.39 ng/mL, respectively.

### 2.2. SFPE Procedure of Sample Preparation

#### 2.2.1. Preliminary Research—SFPE Procedure Optimization

An essential advantage of the SFPE procedure for the preparation of biological samples is the ability to isolate the investigated substances from the matrix components with higher and lower adsorption energy than the test substance and IS [16,18]. To make it possible, in the fourth stage of the procedure (after the step of elution of the test substances from the starting spot—see Section 3.5.3), a chromatographic system should be used, in which the test substances will have a retardation factor (Rf) value close to 0.5 (see: Materials and Methods section). For this purpose, relationships between the retardation factor and the mobile phase composition were investigated. The adsorbents selection for this research stage was according to our previous work [17]. Thus, HPTLC silica gel 60 F254 chromatographic plates were used for the first and second groups of substances. At the same time, HPTLC diol F254 plates were applied for the third substance group. Finally, methanol and toluene mixtures were combined to form the mobile phase. Based on the results from the preliminary study stage [17], it was noticed that using pure methanol as the eluent resulted in very high retardation factor values, often close to or equal to 1. In turn, in systems with pure toluene, all antihypertensive drugs showed very high retention (R_f_ = 0). Consequently, mixing appropriate volumes of a solvent with a relatively high elution strength (methanol) with a solvent with a low elution strength (toluene) will create a mobile phase of elution strength which separates the matrix components of different (lower and higher) adsorption energies from the investigated substances. 

The methanol in toluene solutions were prepared and used as the mobile phases in the following concentrations: 5%, 10%, 30%, 50%, 70%, and 90%. The results are presented as the plots of solute Rf versus methanol content in the mobile phase in Figure 1, Figure 2 and Figure 3.

Considering the first and third groups of drugs, the system in which the retardation factor of the tested substances ranges approximately 0.5 is for the mobile phase solution consisting of 30% methanol in toluene; while for the second group to reach the same Rf value, a system containing 50% methanol in toluene is used. Thus, in such a way, the mobile phase compositions were selected for the fourth stage of the procedure SFPE. Finally, in the fifth stage of the procedure, for the elution of solutes from the adsorbent, methanol (for the second and third groups of substances) or a 0.1% solution of formic acid in methanol (for the first group of substances) was selected (details in our previous work) [17]. Using these mobile phases allowed us to obtain the investigated compounds and the IS in the final solvent front position (see Section 3.5.3), which is accordance with the assumptions of the SFPE procedure [l7].

#### 2.2.2. SFPE—Quantitative Analysis

After selecting the best chromatographic system for the studied group of drugs, meeting the requirements of the SFPE approach, clinical samples containing antihypertensive drugs were prepared for quantitative analysis. To determine the concentration of selected drugs in these samples, it is necessary to plot calibration curves. Calibration curves on the reference plasma were prepared for this purpose, similar to the before-mentioned precipitation method. The concentration ranges used were the same as for the precipitation method. Initially, plasma samples were dispensed onto the chromatography plate. The chromatograms were then developed with the new prototype device according to the procedure described in the ‘’Materials and Methods’’ section. Using the prototype horizontal chamber with a 3D mechanism with a controlled eluent flow allowed for the simultaneous preparation of several samples for further quantitative analysis. This process requires minimal involvement in manual operations. 

After concentrating the substances of interest in the solvent’s final front position, they were extracted into vials from the adsorbent layer. This stage of the procedure was carried out using the CAMAG TLC-MS interface. Next, the vials were placed into the autosampler of the HPLC device; then, the LC-MS/MS analysis was performed. Linear regression equations for the relationships between the substance of interest/IS peak area ratios and the relative concentration of the ratio of analyte/IS are presented in Table 3.

It can be observed in Table 3 that the relationship between the detector response vs. the solute concentration is linear across the whole measurement range for each drug. The determination coefficient (R^2^) values are slightly lower than those obtained by the precipitation method, but they are still acceptable. The best fit of the simple linear regression to the data was obtained for ramipril (R^2^ = 0.998). For valsartan, a very good fit was also obtained, and the coefficient of determination R^2^ = 0.994.

An important aspect in assessing the effectiveness of a given method is the matrix effect; i.e., the effect on the quantification of all other components of the sample except the analyte to be analyzed. Therefore, in the next step of the research, prior to the proper determination of the analyzed drugs in a clinical sample, experiments were carried out to determine the impact of this effect on the obtained results. The studies were performed for each group of drugs by comparing the slope of a calibration curve for standard solutions with that of matrix-matched standard solutions. A lower slope for matrix-matched standard solutions suggests ion suppression, while a higher slope indicates ion enhancement. The matrix effect is shown in the Figure 4, Figure 5 and Figure 6.

There is no significant difference in the slope of the calibration curves for lercanidipine, hydrochlorothiazide and nebivolol. In the case of valsartan, ion-suppression can be observed, while ion enhancement is noticed for ramipril and perindopril.

In the next step, the quantitative analysis of clinical samples was performed. They were prepared in the same way as samples for the standard curves. The three groups of samples of the investigated substances were prepared in the same way as described above, considering the types of stationary and mobile phases selected in the optimization stage of the SFPE procedure. The drug concentrations (ng/mL) in the clinical sample were determined in the same way as for the precipitation method, i.e., by measuring the peak area ratio of the analyte/IS. %RSD, LOD and LOQ were also calculated. In addition, the recovery of the substance from the biological matrix was examined, for the lowest and highest concentration level, in order to determine possible losses of the analyte during the SFPE procedure, thus determining the accuracy of the method. Additionally, the precision of the method was estimated by calculating the coefficient of variation (CV). Intra-day and inter-day (three consecutive days) precision was assessed by analyzing five replicates at three different substance concentration levels, i.e., C_1_, C_3_, C_5_. The results of the quantitative analysis for the investigated drugs are presented in Table 4. 

The results obtained with the SFPE technique (Table 4) are close to the values obtained using the precipitation sample preparation technique (Table 2) (with the exception of ramipril). Quantitative determination of all solutes was at a satisfactory level. The %RSD of the results obtained by the LC-MS/MS method in combination with the SFPE procedure fluctuates around 5%. This is acceptable and shows that the prototype equipment used to prepare the clinical samples with the investigated substances gives reproducible results. Even though the LOD and LOQ values from SFPE are higher than the precipitation procedure, but they are still satisfactory. The recovery of the analyzed drugs is in the range of 79.88–120.36%. When determining trace amounts of a substance, recovery at this level is acceptable [31], so the SFPE procedure can be considered accurate. Moreover, low values of the coefficient of variation (CV) for most drugs prove the high precision of the method.

Since the results of the determination of selected antihypertensive drugs are comparable in the two discussed procedures, it can be concluded that the SFPE technique is as good for the preparation of samples for quantitative analysis as the reference method. Therefore, a correlation analysis was performed to assess the compatibility between these two methods. For this purpose, a correlation plot (Figure 7) was made. We compare concentrations of the test substances obtained after using precipitation (ordinate axis) and SFPE (abscissa axis) as a sample preparation method. As can be seen, there is a strong relationship between the variables (correlation coefficient close to 1) in the investigated concentration range. This indicates that SFPE can replace the precipitation method.

Nevertheless, to determine which mode is more economical and effective, it is necessary to compare them in terms of the preparation time of a single sample and solvent consumption. Since the similar problem we discussed in our previous work [17], we have chosen the same concepts. Thus, assuming the maximum dimensions of the chromatography plate (i.e., 10 × 20 cm), the preparation time of a single sample for quantification with the SFPE procedure would be 2.40 min. Moreover, the required amount of solvent is 0.38 mL [17]. The precipitation technique involves 24 samples (24 samples could be centrifuged simultaneously), which would be 2.5 min. [16]. Considering solvent consumption, the precipitation of proteins and the extraction of metabolites from the biological matrix requires three times more solvent than the plasma volume [16]. Hence, the solvent consumption will be 0.3 mL. Therefore, using the SFPE technique instead of the protein precipitation procedure before LC-MS/MS analysis does not significantly reduce the time or solvent consumption. However, a definite advantage of our approach is better specificity, which allows selective extraction of only specific analytes. Conversely, precipitation can result in the precipitation of both the target analyte and other components from the matrix. Moreover, an additional benefit of SFPE is the ability to automate the procedure thoroughly. It would allow for increased efficiency, repeatability, and reduction of manual operations. Research is currently being conducted in this direction.

## 3. Materials and Methods

### 3.1. Materials and Reagents

Antihypertensive drugs: ramipril, lercanidipine, valsartan, hydrochlorothiazide, telmisartan, perindopril, and nebivolol were purchased from Sigma-Aldrich (St. Louis, MO, USA). The structures, physicochemical properties and drug classes (pharmacological properties) of investigated substances are presented in Table 5. Chromatographic glass plates, HPTLC silica gel 60 F254 and HPTLC diol F254 were supplied by Merck (Darmstadt, Germany). Acetonitrile and methanol (both MS purity) were purchased from Fisher Chemical (Waltham, MA, USA). Ethanol, toluene and formic acid (98–100%) LC-MS grades were supplied by Merck (Darmstadt, Germany). The ultra-pure water was obtained from the Millipore Direct-Q3-UV purification system (Merck, Darmstadt, Germany). The plasma samples (also a reference sample) were kindly provided by the Cardiology Clinic of the Independent Public Clinical Hospital No. 4 in Lublin (Lublin, Poland). 

### 3.2. Devices and Instrumentation

The following devices and instruments were used in this research: vortexer P/N: 541-11000-00-1 (Heidolph, Schwabach, Germany), analytical balance XPR6UD5 (Mettler-Toledo Greifensee, Switzerland), centrifuge MPW-351 R (Medipment, Warsaw, Poland), Eppendorf Research Plus pipette set (Eppendorf AG Hamburg, Germany), TLC plate cutter (OM Laboratory, Chigasaki, Japan), laboratory dryer Pol-Eko 115 SLW (Pol-Eko-Aparatura, Wodzisław Śląski, Poland), horizontal thin-layer chromatography chamber DS-II-10×10 (Chromdes, Lublin, Poland), CAMAG TLC visualiser with WinCATS software for plate image documentation of the solute zones onto the adsorbent layer (CAMAG, Muttenz, Switzerland), a prototype horizontal chamber with a moving pipette driven by 3D mechanism (Infinum 3D, Lublin, Poland), CAMAG TLC-MS Interface for the extraction of substances from the adsorbent surface (CAMAG, Muttenz, Switzerland). The Agilent 1290 Infinity II LC System (Agilent Technologies, Santa Clara, CA, USA) connected with the Agilent 6470 Triple Quadrupole was used for the LC-MS (Agilent Technologies, Santa Clara, CA, USA) experiments. 

### 3.3. Standard Solutions

Stock solutions of the drugs and stock solutions of the IS were prepared by dissolving the appropriate amounts of standards in methanol. The stock solution concentration was 1 mg/mL for each antihypertensive drug and each IS. The obtained solutions were stored in a refrigerator at 4 °C. 

### 3.4. Calibration Curve

Samples for calibration were prepared by adding appropriate amounts of stock solutions and IS to 200 μL of reference plasma. Calibration curves were designed for three groups of substances. A different IS has been added to each group. The final concentrations of antihypertensive drugs in the plasma are presented in Table 6. Concentration ranges were selected based on therapeutic drug concentrations, and the latter ones are shown in Table 5.

### 3.5. Plasma Sample Preparation

#### 3.5.1. Common Stage of Experiments

Plasma samples (also the reference sample) provided by the Cardiology Clinic were stored at −80 °C until analysis. Samples obtained from three patients with apparent difficult-totreat hypertension were assessed (each patient took a different group of drugs). Then, the frozen plasma samples were thawed at a temperature of 4 °C. Reference plasma samples were used to prepare calibration curves (details in Section 3.4). In turn, in the case of the suitable samples, the IS was added to them (different for each group of substances). The samples prepared in this way were divided into two parts. Each was subjected to quantitative analysis using two different protocols: precipitation and SFPE.

#### 3.5.2. Precipitation Method 

Protein precipitation and the extraction of target substances were performed by adding 300 μL of a cold (−20 °C) mixture of methanol and ethanol (1:1 *v*/*v*) to 100 μL of plasma [41,42,43]. The samples were vortexed for 30 seconds and then cooled for 15 minutes at −20 °C. The use of cold denaturation additionally destabilizes the structure of the protein [44,45]. After this, the pellet was removed by centrifugation at 16,000× *g* for 10 min at 4 °C. The supernatant was retrieved from above the pellet and filtered through a nylon filter (0.2 μm) into inserts. The solutions obtained were quantified by LC-MS/MS analysis. 

#### 3.5.3. SFPE Procedure 

Chromatographic glass plates, HPTLC silica gel 60 F254, and HPTLC diol F254 were cut into smaller pieces (5 × 10 cm) with a TLC plate cutter. Before chromatogram development, the plates were pre-washed in methanol for one minute, dried in the air, and then activated at 105−110 °C for 15 minutes in a laboratory dryer. Next, the plates were left in a desiccator for cooling. Then, the samples were applied onto the prepared chromatography plates as tiny single drops (six spots: including one blank) in a volume of 3 μL by using the automatic pipette at a distance of 15 mm from the lower edge of the chromatography plate (starting line) (Figure 8A). Then, the plate was left for 10 minutes in the fume hood to dry the spots of the sample. 

After evaporation of the solvent, the plate was placed in the prototype horizontal chamber for TLC with a 3D mechanism. The starting spots were wetted twice with the mobile phase before the first development of the chromatogram. The process of wetting the zones of substances is described in detail in our previous article [17]. During chromatogram development, the movement path of the pipette delivering the mobile phase to the adsorbent layer was parallel to the x-axis. At the same time, its position toward the y-axis was constant (Y1 = 10 mm from the lower edge of the plate) (Figure 8A). The speed of the pipette movement was set to 2000 mm/min, while the mobile phase delivery rate was 6 mL/h. The distance between the pipette tip and the surface of the adsorbent layer was 0.15 mm. The solvents used for developing the chromatograms were selected at the optimization stage of the SFPE procedure (details in Section 2.2.1). The first chromatogram was developed with methanol or 0.1% formic acid in methanol (depending on the substance group) at a distance of 10 mm, measured from the starting line (Figure 8B). Then, after evaporation of the solvent (10 min), the second chromatograms of samples were developed with a mixture of toluene and methanol (details in the section optimization) at a distance of 25 mm (Figure 8C) and allowed to dry for 10 min. Based on the optimization of the SFPE procedure, the test antihypertensive drugs were noticed to reach a distance of approximately 20 mm (visualization of substance zones using CAMAG TLC Visualizer) (Figure 8C). Hence, the subsequent development of the chromatogram (third) was carried out over a distance of 22 mm, so that the zones of the substances of interest (also zones of the IS) were focused on the solvent front position— the red dotted line (Figure 8D). At this stage, the pipette was set to the Y2 = 30 mm position, measured from the edge of the chromatography plate; methanol or 0.1% formic acid in methanol was used as the mobile phase (see optimization section for details). Subsequently, after evaporating the solvent from the adsorbent layer, the substances were extracted from the solvent front position using the TLC-MS interface into a 150 µL insert vial and subjected to LC-MS/MS analysis. The extracting solvent was methanol or 0.1% formic acid in methanol, while the flow velocity of the solvent was 0.3 mL/min.

Figure 8 only provides a detailed description of one of the stages of the SFPE procedure, namely the development of the chromatograms. In turn, the graphical presentation of the entire procedure was presented in our previous work [17].

### 3.6. LC-MS/MS Analysis Conditions

For the first, second and third groups of substances, an Agilent 1290 Infinity II LC System coupled with an Agilent 6470 Triple Quadrupole was used. The chromatography (classical and SFPE procedures) was performed with the Zorbax Eclipse Plus C18 column (RRHD 2.1 × 50 mm, 1.8 μm). The mobile phase consisted of solvent A (0.1% formic acid in water) and solvent B (100% acetonitrile). The gradient elution was conducted as follows: 0 min: 95% A, 5% B; 8 min: 5% A, 95% B, for the first and second groups of substances; the analysis run time was 8 min. Conversely, for the third group, it was as follows: 0 min: 95% A, 5% B; 6.50 min: 60% A, 40% B; 6.51 min: 5% A, 95% B; 8 min: 5% A, 95% B; the analysis run time was 8 min. The flow rate was 0.4 mL/min. MS data were obtained in the positive ion modes (multiple reaction monitoring mode) with electrospray probe voltages of 4000 V. The nebulizer gas setting was 45 psi for the first group of substances and 40 psi for the second and third. The ion source was operated at a temperature of 350 °C and a drying gas setting of 12 L/min for the first and third group substances, and 250 °C, 10 L/min for the second group. The MRM (multiple reaction monitoring) transitions for each group of substances are shown in Table 7.

### 3.7. Ethics Approval and Consent to Participate

Clinical samples were collected after obtaining informed consent from study participants by the guidelines of the Bioethics Committee of the Medical University of Lublin (number KE-0254/54/03/2022, 24.03.2022, Lublin, Poland). The method was tested on patients with suspected refractory arterial hypertension treated with standard doses of antihypertensive drugs in combination regimens.

## 4. Conclusions

Our innovative procedure for the sample preparation by SFPE allowed us to isolate the substances of interest from the biological matrix successfully. The concentrations of the tested drugs are very similar to the results obtained with the reference method. In addition, acceptable values of validation parameters were obtained, which proves the effectiveness of the SFPE procedure for the preparation of biological samples. The preparation time of a single sample and solvent consumption is very satisfactory. Moreover, the high selectivity method and ease of full automation makes the currently used methods of sample preparation of biological origin very competitive. Additionally, the presented approach may be an effective tool in monitoring resistant arterial hypertension pharmacotherapy. Nevertheless, research on a broader drug group should be done to apply the method comprehensively.

## Figures and Tables

**Figure 1 molecules-28-02213-f001:**
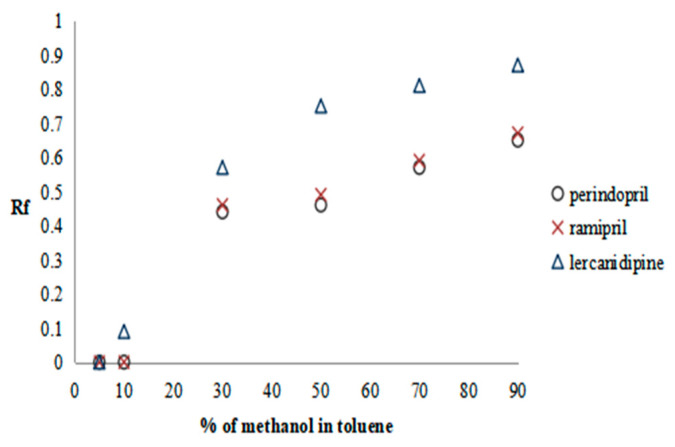
Plots of substance retention relationships vs. the mobile phase composition for the first group of drugs. Stationary phases: HPTLC silica gel 60 F_254_.

**Figure 2 molecules-28-02213-f002:**
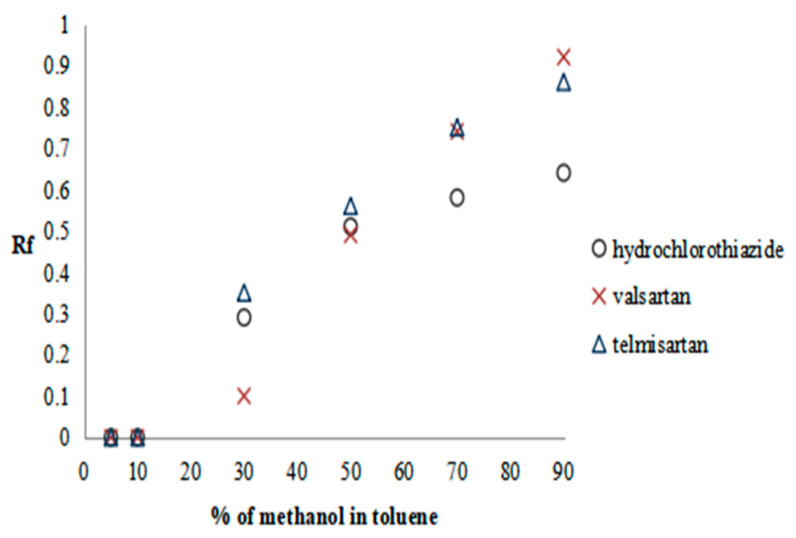
Plots of substance retention relationships vs. the mobile phase composition for the second group of drugs. Stationary phases: HPTLC silica gel 60 F_254_.

**Figure 3 molecules-28-02213-f003:**
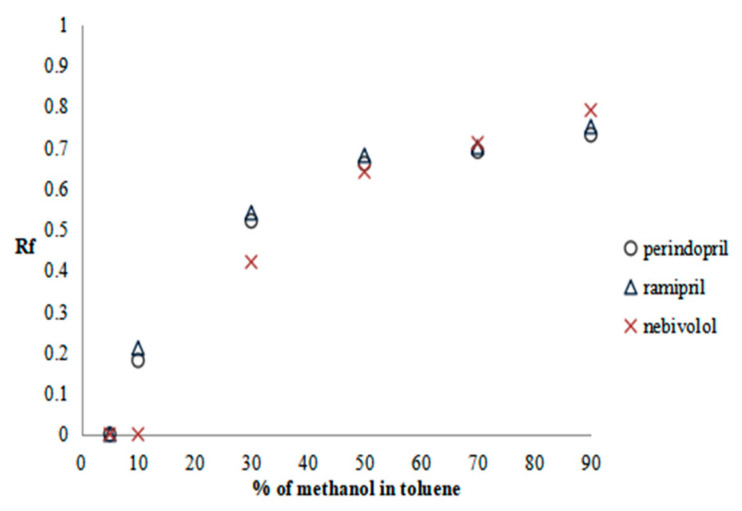
Plots of substance retention relationships vs. the mobile phase composition for the third group of drugs. Stationary phases: HPTLC diol F_254_.

**Figure 4 molecules-28-02213-f004:**
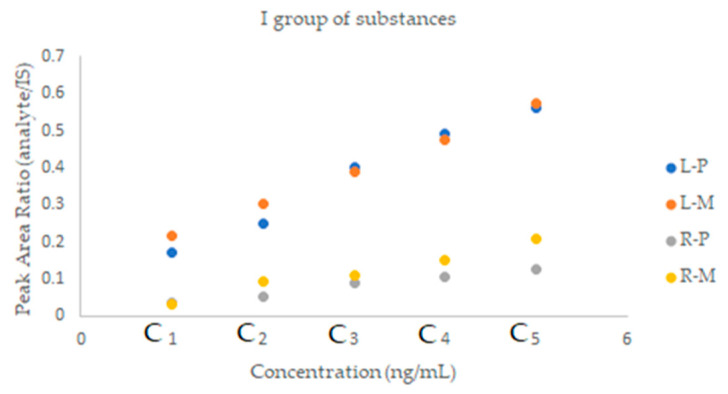
A graphical representation of the matrix effect between two parallel calibration curves for the I group substances, prepared without and with the matrix. P—samples without the matrix, M—samples with the matrix.

**Figure 5 molecules-28-02213-f005:**
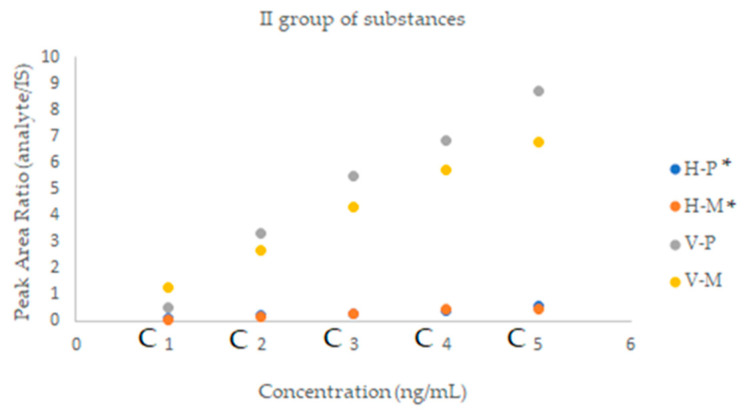
A graphical representation of the matrix effect between two parallel calibration curves for the II group substances, prepared without and with the matrix. P—samples without the matrix, M—samples with the matrix; *—value multiplied by hundred (for better visibility).

**Figure 6 molecules-28-02213-f006:**
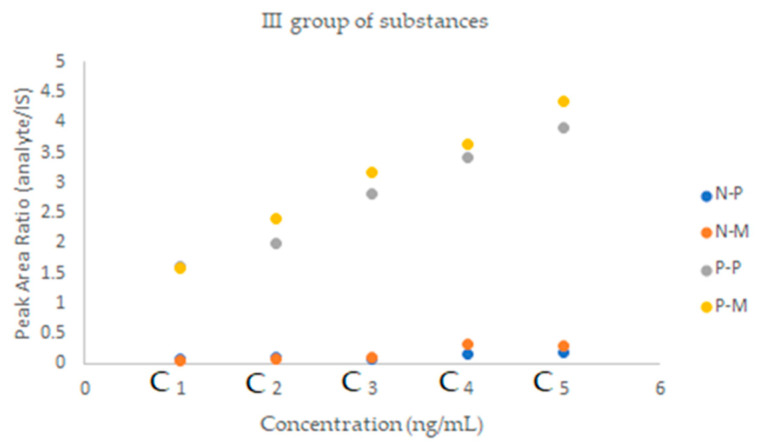
A graphical representation of the matrix effect between two parallel calibration curves for the III group substances, prepared without and with the matrix. P—samples without the matrix, M—samples with the matrix.

**Figure 7 molecules-28-02213-f007:**
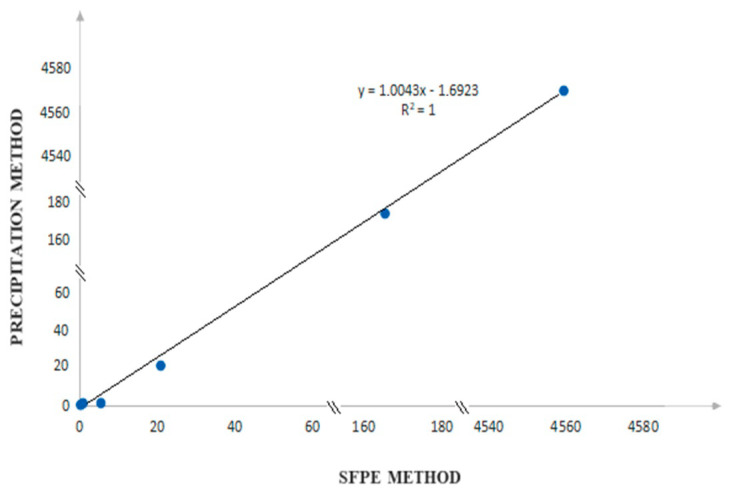
Correlation analysis between the SFPE procedure and the reference method.

**Figure 8 molecules-28-02213-f008:**
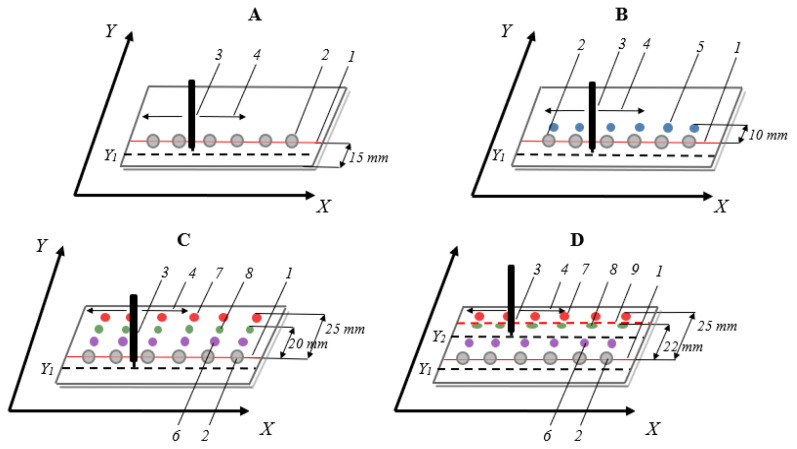
Stages of biological sample preparation by solvent front position extraction (details in the text). The dashed lines mark the path of the pipette delivering the eluent. 1—starting line, 2—starting spots of the samples, 3—pipette delivering the eluent, 4—the direction of movement of the pipette, 5—substances eluted from the plasma zone, 6—matrix components of higher retention than the substances of interest, 7—matrix components of lower retention than the substances of interest, 8—the zone of substances of interest (including the IS), 9—a line indicating the appropriate (final) front of the mobile phase. (**A**)—initial position of the pipette before chromatogram development; (**B**)—first, short distance elution of the test substances from starting spot; (**C**)—second elution of the substances to reach R_f_ values close to 0.5; (**D**)—third elution to reach the solvent front position of the investigated substances. For details, see the text.

**Table 1 molecules-28-02213-t001:** Linear regression equations for the test drugs determined with precipitation procedure.

Substance of Interest	Linearity	R^2^
Lercanidipine	y = 1.111x − 0.004	0.999
Ramipril	y = 1.226x + 0.022	0.989
Hydrochlorothiazide	y = 0.01x + 2.821	0.998
Valsartan	y = 0.833x + 0.907	0.998
Nebivolol	y = 0.169x − 0.002	0.997
Perindopril	y = 0.840x − 0.018	0.998

**Table 2 molecules-28-02213-t002:** The figures of merits of the precipitation procedure, calculated on the basis of the average (n = 5). Results obtained for three patients with resistant hypertension (each patient took a different group of drugs). %RSD—relative standard deviation; LOD—limit of detection, LOQ—limit of quantitation.

I Group of Substances *				
	Concentration (ng/mL)	%RSD	LOD (ng/mL)	LOQ (ng/mL)
Lercanidipine	0.16	0.44	0.02	0.05
Ramipril	1.11	6.48	0.32	0.97
**II Group of Substances ****				
Hydrochlorothiazide	165.56	2.05	0.63	1.90
Valsartan	4577.62	0.52	4.42	13.39
**III Group of Substances *****				
Nebivolol	1.00	4.84	0.12	0.36
Perindopril	19.49	2.50	0.47	1.44

* perindopril used as the IS; ** telmisartan used as the IS; *** ramipril used as the IS and calculated based on the average value. The abbreviation IS—respectively, the internal standard.

**Table 3 molecules-28-02213-t003:** Linear regression equations for the test drugs determined with the SFPE (solvent front position extraction) procedure.

Substance of Interest	Linearity	R^2^
Lercanidipine	y = 1.286x − 0.008	0.981
Ramipril	y = 1.109x − 0.012	0.998
Hydrochlorothiazide	y = 0.063x + 0.014	0.984
Valsartan	y = 3.275x − 2.508	0.994
Nebivolol	y = 0.241x − 0.002	0.993
Perindopril	y = 1.543x − 0.673	0.981

**Table 4 molecules-28-02213-t004:** The figures of merits of the SFPE procedure, calculated on the basis of the average (n = 5). Results obtained for three patients with resistant hypertension (each patient took a different group of drugs). The solute recovery is calculated from the formula Mx/dLC-MS × 100, where Mx- is the value of the peak area substance obtained with the use of LC-MS/MS method after sample preparation using the SFPE technique; dLC-MS is the value of the peak area substance in the standard solution obtained with the use of LC-MS/MS method without the use of the SFPE technique. %CV—coefficient of variation is calculated from the formula SD/Mean × 100, where SD is the standard deviation.

I Group of Substances *							
	Conc. (ng/mL)	%RSD	LOD (ng/mL)	LOQ (ng/mL)	%Recovery	%CV Intra-day	%CV Inter-day
Lercanidipine	0.19	5.41	0.06	0.17	98.44–105.01	6.61–1.49	5.22–1.21
Ramipril	5.43	0.92	0.16	0.47	117.63–84.36	6.04–2.61	5.30–2.11
**II Group of Substances ****							
Hydrochloro thiazide	166.41	5.84	1.46	4.41	88.90–79.88	7.45–4.58	9.74–7.97
Valsartan	4559.76	4.97	9.78	29.64	107.20–115.40	5.04–1.87	5.60–2.40
**III Group of Substances *****							
Nebivolol	0.83	4.75	0.23	0.70	120.36–80.92	7.70–1.85	6.44–3.13
Perindopril	22.10	1.73	1.82	5.52	108.26–85.42	4.14–1.34	3.64–1.10

* perindopril used as the IS; ** telmisartan used as the IS; *** ramipril used as the IS and calculated based on the average value. The abbreviation IS—respectively, the internal standard.

**Table 5 molecules-28-02213-t005:** A short description of the substances studied.

Name of the Substance	Formula	Value pK_a_ *	Concentration Ranges Therapeutic (ng/mL)	Drug Classes
lercanidipine	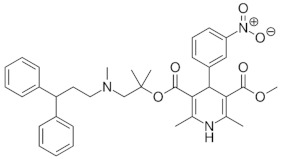	9.36 (ionogenic basic group)	1.23–10.23	calcium channel blockers (CCB)
ramipril	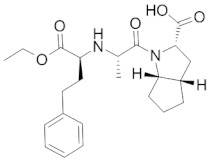	3.75 (ionogenic acid group) 5.2 (ionogenic basic group)	11–35	angiotensin converting enzyme (ACE) inhibitors
hydrochlorothiazide	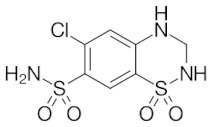	9.09 (ionogenic acid group)	70–376	thiazide diuretic
valsartan	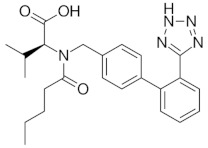	4.35 (ionogenic acid group)	1940–6400	angiotensin II receptor blockers (ARB)
telmisartan	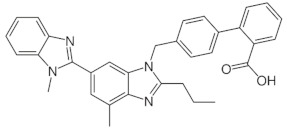	3.62 (ionogenic acid group) 5.86 (ionogenic basic group)	8.9–366	angiotensin II receptor blockers (ARB)
nebivolol	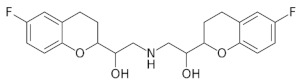	8.9 (ionogenic basic group)	1.48–26.1	β-blockers
perindopril	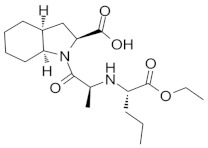	3.79 (ionogenic acid group) 5.48 (ionogenic basic group)	20 – 83	angiotensin converting enzyme (ACE) inhibitors

* pK_a_—negative logarithm of the equilibrium constant K of reversible chemical reactions, where Ka—is the acid dissociation constant, all data taken from [32,33,34,35,36,37,38,39,40].

**Table 6 molecules-28-02213-t006:** Calibration curve concentrations of individual antihypertensive drugs and IS in the human plasma (ng/mL).

Calibration Level					
Substance	1	2	3	4	5
Group I					
ramipril	10	20	30	40	50
lercanidipine	1	2	5	7.5	10
perindopril (IS) *	40	40	40	40	40
Group II					
valsartan	1500	3000	4500	5500	6500
hydrochlorothiazide	60	140	220	300	380
telmisartan (IS) **	200	200	200	200	200
Group III					
perindopril	15	30	50	70	90
nebivolol	1	5	10	20	30
ramipril (IS) ***	20	20	20	20	20

* perindopril used as the IS; ** telmisartan used as the IS; *** ramipril used as the IS. The abbreviation IS—respectively, the internal standard.

**Table 7 molecules-28-02213-t007:** (Multiple reaction monitoring (MRM) transitions.

Substance	Prec. Ion (*m/z*)	Prod. Ion (*m/z*)	Frag. (V)	CE (eV)	Polarity	Ret. Time (min)
Lercanidipine	612.3	280.1/100	145	26/40	Positive	4.94
Ramipril	417.2	253.8/234	128	26/22	Positive	3.75 */6.03 **
Perindopril	369.2	172/98	103	22/35	Positive	3.33 */5.19 **
Hydrochlorothiazide	298	268.8/204.9	96	10/18	Positive	2.1
Valsartan	436.2	235/207	83	18/30	Positive	5.39
Telmisartan	515.2	497/276	155	35	Positive	3.98
Nebivolol	406.2	150.9/123	112	35	Positive	6.44

* analysis of the first group of substances, ** analysis of the third group of substances.

## Data Availability

All obtained data was presented in the manuscript.

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
