# Peer review of "Comparison of the Determination of Some Antihypertensive Drugs in Clinical Human Plasma Samples by Solvent Front Position Extraction and Precipitation Modes"

_molecules, 2023, doi:10.3390/molecules28052213_

Round 1
Reviewer 1 Report
Comments
The article entitled “Comparison of the determination of some antihypertensive drugs in clinical human plasma samples by solvent front position extraction and precipitation modes” by JagliÅ„ska et al., provides an interesting perspective on effective sample preparation and analysis of antihypertensive drugs using HPLC-MS. The author has presented a well-organized experimental section along with research findings and makes a valuable contribution to the existing body of knowledge in analytical area. I have a few minor suggestions in the manuscript and recommend the article for publication after revisions.
Lines 24-25: Results obtained by SFPE were very satisfactory (linearity R2 ≥ 0.981, %RSD ≤ 6%, LOD and LOQ were in the range of 0.1554 ng/L–9.7808 ng/L and 0.4710 ng/L–29.6388 ng/L, respectively).
Q. LOD and LOQ values can be limited to two decimal points instead of four decimals.
Example 0.155 ng/L – 9.78 ng/L
Lines 24-25: The calibration curves for every antihypertensive drug have a coefficient of determination close to 1.
Based on Table 1, the above sentence should be revised to-
The calibration curves developed for antihypertensive drugs have a coefficient of determination (r2) value in the range of 0.997 to 0.999.
Line 123-Table 2: Present all the values up to two decimal places.
Analyte concentrations were given in ng/L (example: Lercanidipine 0.163 ± standard deviation?). Despite this, there is no information on how many samples were analyzed (duplicates or triplicates) and what is the standard deviation within the group analysis. In order to demonstrate the efficiency of your analytical method, please provide more information, including standard deviation values
Line 130-131: All values should be uniform up to two decimal places throughout the manuscript.
Line 366-Table 6: Collision cell energies are measured in electron volts (eV). Revise units in the Table. Is 35 eV the optimal collision cell energy for the MRM transition ions of Telmisartan (497/276) and Nebivolol (150.9/123)?
Author Response
The article entitled “Comparison of the determination of some antihypertensive drugs in clinical human plasma samples by solvent front position extraction and precipitation modes” by JagliÅ„ska et al., provides an interesting perspective on effective sample preparation and analysis of antihypertensive drugs using HPLC-MS. The author has presented a well-organized experimental section along with research findings and makes a valuable contribution to the existing body of knowledge in analytical area. I have a few minor suggestions in the manuscript and recommend the article for publication after revisions.
- Lines 24-25: Results obtained by SFPE were very satisfactory (linearity R2 ≥ 0.981, %RSD ≤ 6%, LOD and LOQ were in the range of 0.1554 ng/L–9.7808 ng/L and 0.4710 ng/L–29.6388 ng/L, respectively).
- LOD and LOQ values can be limited to two decimal points instead of four decimals.
Example 0.155 ng/L – 9.78 ng/L
Answer: It has been changed as suggested.
- Lines 24-25: The calibration curves for every antihypertensive drug have a coefficient of determination close to 1.
Based on Table 1, the above sentence should be revised to-
The calibration curves developed for antihypertensive drugs have a coefficient of determination (r2) value in the range of 0.997 to 0.999.
Answer: Good point. It has been changed (see lines: 128-129).
- Line 123-Table 2:Present all the values up to two decimal places.
Answer: It has been changed.
- Analyte concentrations were given in ng/L (example: Lercanidipine 0.163 ± standard deviation?). Despite this, there is no information on how many samples were analyzed (duplicates or triplicates) and what is the standard deviation within the group analysis. In order to demonstrate the efficiency of your analytical method, please provide more information, including standard deviation values.
Answer: Good point. Information is supplemented in the text (see Table 2, 4).
- Line 130-131: All values should be uniform up to two decimal places throughout the manuscript.
Answer: It was done.
- Line 366-Table 6: Collision cell energies are measured in electron volts (eV). Revise units in the Table. Is 35 eV the optimal collision cell energy for the MRM transition ions of Telmisartan (497/276) and Nebivolol (150.9/123)?
Answer: Good point. It has been changed (see Table 7).
Yes, 35 eV is the optimal collision cell energy for the MRM transition ions of telmisartan (497/276) and nebivolol (150.9/123), selected by optimization.

Reviewer 2 Report
The presented manuscript disscuss the new mode of sample preparation, solvent front position extraction (SFPE) in comparison with precipitation mode used during determination of some antihypertensive drugs in clinical human plasma samples. The authors mention that preliminary research on the suitability of SFPE for the preparation of biological samples with antihypertensive drugs before their further quantitative determinations were described in previous paper (Molecules 2022, 27, 205. https://doi.org/10.3390/molecules27010205.) For this reason, the authors' statement that “ The SFPE procedure was used for the first time to prepare a clinical sample containing the drugs mentioned above ….” Is somewhat debatable. Overall, the study is conducted correctly and successfully. The provided paper is a topic of Molecules and can be accepted for publishing after the following minor revisions:
1. It is not clear what is the criterion for dividing the tested drugs into three groups of substances (see table 2)
2. Table 2 shows the concentration of the analyzed substances and the %RSD value. How many repetitions of each determination were made. Please enter the n value into the table.
3. Line 201. I disagree with the statement that R2 is outstanding in all cases. The acceptance criterion for R2 according to the ICH recommendation is 0.995. Thus, it is fully met only for ramipril (0.998) and almost met for valsartan (0.994).
4. Table 4 – similarly to Table 2, please enter the n value.
5. Lines 214-215. Authors stated: “The results obtained with the SFPE technique (Table 4) are close to the values obtained using the precipitation sample preparation technique (Table 2)” it is not true in the case ramipril. Please, discuss.
6. Lines 285-286. It would be good to provide information about the therapeutic ranges of the tested drugs in human plasma. This would make it possible to assess the suitability of the presented method for analyzes in clinical samples.
7. Line 300. Are the analyzed drugs bound to plasma proteins? If so, to what extent? It should be noted that part of the drug could be removed of in the protein precipitation step if the hydrolysis step is not introduced. Please discuss it. Moreover, in the same statement authors stated: “extraction of metabolites”. What were the metabolites?
8. Section 3.7. For complete correctness, the number and year of approval of the Bioethics Committee for the study should be provided.
9. Conclusion (line 375). The authors state that the results obtained using the SFPE method are very similar to the results obtained using the classical method (protein precipitation only). So what is the added benefit of this approach?
Author Response
The presented manuscript disscuss the new mode of sample preparation, solvent front position extraction (SFPE) in comparison with precipitation mode used during determination of some antihypertensive drugs in clinical human plasma samples. The authors mention that preliminary research on the suitability of SFPE for the preparation of biological samples with antihypertensive drugs before their further quantitative determinations were described in previous paper (Molecules 2022, 27, 205. https://doi.org/10.3390/molecules27010205.) For this reason, the authors' statement that “ The SFPE procedure was used for the first time to prepare a clinical sample containing the drugs mentioned above ….” Is somewhat debatable. Overall, the study is conducted correctly and successfully. The provided paper is a topic of Molecules and can be accepted for publishing after the following minor revisions:
- It is not clear what is the criterion for dividing the tested drugs into three groups of substances (see table 2).
Answer: Good point. Information is supplemented in the text (see lines: 132-133).
- Table 2 shows the concentration of the analyzed substances and the %RSD value. How many repetitions of each determination were made. Please enter the n value into the Table.
Answer: The information has been supplemented.
- Line 201. I disagree with the statement that R2 is outstanding in all cases. The acceptance criterion for R2 according to the ICH recommendation is 0.995. Thus, it is fully met only for ramipril (0.998) and almost met for valsartan (0.994).
Answer: Good point. It has been changed (see lines: 270-272).
- Table 4 – similarly to Table 2, please enter the n value.
Answer: It was done.
- Lines 214-215. Authors stated: "The results obtained with the SFPE technique (Table 4) are close to the values obtained using the precipitation sample preparation technique (Table 2)" it is not true in the case ramipril. Please, discuss.
Answer: At this research stage, it is difficult to say what cause the difference in the case of ramipril. Nevertheless, please note that the SFPE procedure is a new technique that is constantly being improved. Further research in a broader group of drugs is planned, which may give us an answer. Nevertheless, to avoid interpretation controversies the statement in the manuscript was changed into: “The results obtained with the SFPE technique (Table 4) are close to the values obtained using the precipitation sample preparation technique (Table 2) (with exception for the ramipril).
- Lines 285-286. It would be good to provide information about the therapeutic ranges of the tested drugs in human plasma. This would make it possible to assess the suitability of the presented method for analyzes in clinical samples.
Answer: The information has been supplemented. Therapeutic ranges are given in Table 5.
- Line 300. Are the analyzed drugs bound to plasma proteins? If so, to what extent? It should be noted that part of the drug could be removed of in the protein precipitation step if the hydrolysis step is not introduced. Please discuss it. Moreover, in the same statement authors stated: "extraction of metabolites". What were the metabolites?
Answer: Good point. The hydrolysis step was omitted, but we introduced the data on cold denaturation. Information was supplemented in the text (see the line: 375). As for the statement "extraction of metabolites," the wording is wrong. It has been replaced by "target substances" (see the line: 372).
- Section 3.7. For complete correctness, the number and year of approval of the Bioethics Committee for the study should be provided.
Answer: The information has been supplemented.
- Conclusion (see line 375). The authors state that the results obtained using the SFPE method are very similar to those obtained using the classical method (protein precipitation only). So what is the added benefit of this approach?
Answer: An additional benefit of our approach is higher selectivity and the possibility of full automation of the procedure (see lines: 454-460).

Reviewer 3 Report
The manuscript submitted described “Comparison of the determination of some antihypertensive drugs in clinical human plasma samples by solvent front position extraction and precipitation modes”. Herein, the author’s performed the determination of the selected antihypertensive drugs in human plasma samples with the novel Solvent Front Position Extraction (SFPE) technique. Moreover, the method successfully applied for the real sample analysis. Overall, the work looks significant advancement to the existed literature but still some concerns need to be resolved. Therefore, I recommend this manuscript for publication after major revision. The following comments need to be considered before submitting.
Comments:
1. The abstract should be more detailed and should highlight the novelty of proposed method.
2. The introduction has missing points like the novelty, advantages of the proposed method, more detailed description about analytes and aim of the study. The introduction should include the names of the hypertensive drugs. The previous studies related to it.
3. In order to get readers visibility authors are suggested to include chemical structures of all drugs including common properties.
4. What are the pKa values of analytes? What was the sample pH? Why any pH optimization study was not done?
5. Why only methanol and ethanol was used for protein precipitation? ACN is also a good choice and used in most of the studies. Please cite the reference for the same or perform the optimization study
6. In table 2 what does concentration refer to? Is it the obtained conc? Is this data only for one patient? How many subjects were involved in the study?
7. What were recoveries? How did the author make sure of the accuracy of the method?
8. What was the analysis run time? The authors must include the chromatograms for real sample analysis.
9. Method validation is not complete inter and intraday precision should be performed.
10. In order to get readers visibility authors are suggested provide graphical representation of SFPE procedure
11. How about the matrix effect of plasma samples? Matrix effect must be calculated
12. In the study, many factors were investigated. But it should be pointed out that the results analysis is quite simple. More in-depth analysis is required.
13. Comparison of the current developed method with other extraction methods should be included.
14. The conclusion part should be revised, author make a clear conclusion highlighting the novel points precisely.
15. There are many typo and grammatical errors in the manuscript. The authors should revise it carefully. English polish is also necessary.
Author Response
The manuscript submitted described “Comparison of the determination of some antihypertensive drugs in clinical human plasma samples by solvent front position extraction and precipitation modes”. Herein, the author’s performed the determination of the selected antihypertensive drugs in human plasma samples with the novel Solvent Front Position Extraction (SFPE) technique. Moreover, the method successfully applied for the real sample analysis. Overall, the work looks significant advancement to the existed literature but still some concerns need to be resolved. Therefore, I recommend this manuscript for publication after major revision. The following comments need to be considered before submitting.
- The abstract should be more detailed and should highlight the novelty of proposed method.
Answer: The abstract has been supplemented with additional information. The novelty of the proposed procedure is described in lines 15-18.
- The introduction has missing points like the novelty, advantages of the proposed method, more detailed description about analytes and aim of the study. The introduction should include the names of the hypertensive drugs. The previous studies related to it..
Answer: This information is included in the text (see lines: 96-105).
- In order to get readers visibility authors are suggested to include chemical structures of all drugs including common properties.
Answer: Good point. The structures and physicochemical properties of investigated substances are presented in Table 5.
- What are the pKa values of analytes? What was the sample pH? Why any pH optimization study was not done?
Answer: The pKa values of the analytes are presented in Table 5.
The primary assumption of the SFPE procedure is to find a chromatographic system that would ultimately allow obtaining the tested compounds and the internal standard in the final position of the solvent front with the smallest possible number of developments. For the second and third groups of substances, methanol was chosen as the mobile phase based on the preliminary research carried out in our previous article. Thus, it was not necessary to perform further pH optimization. In turn, it was carried out for 1 group of substances because preliminary research did not give a clear answer that meets the postulates of the SFPE. Therefore, a 0.1% solution of formic acid in methanol was selected as the mobile phase for the first group of drugs (see lines: 240-245).
- Why only methanol and ethanol was used for protein precipitation? ACN is also a good choice and used in most of the studies. Please cite the reference for the same or perform the optimization study.
Answer: I agree, but the methanol/ethanol mixture is often used in research as an extraction mixture. Literature is supplemented in the text (see the line: 374).
6 In table 2 what does concentration refer to? Is it the obtained conc? Is this data only for one patient? How many subjects were involved in the study?
Answer: Investigated concentration in Table 2 refers to this obtained concentration in the clinical sample (see lines: 133-134). The data are for three patients.
The information has been supplemented in the text (see Tables 2, 4)
- What were recoveries? How did the author make sure of the accuracy of the method?
Answer: The recoveries were determined. Details are presented in section 2.2.2.
- What was the analysis run time? The authors must include the chromatograms for real sample analysis.
Answer: Information on the analysis run time is added in the text (see lines: 616 and 618). Regarding real sample chromatograms, it was considered preferable to add the retention times of the substances (see Table 7).
- Method validation is not complete inter and intraday precision should be performed.
Answer: It was done. Details are presented in section 2.2.2.
- In order to get readers visibility authors are suggested provide graphical representation of SFPE procedure.
Answer: The graphical presentation of the entire procedure SFPE, was presented in our previous work. The citation was added in the text (see the line: 427).
- How about the matrix effect of plasma samples? Matrix effect must be calculated.
Answer: Good point. The matrix effect on the investigated groups of drugs is shown in graphs 4, 5 and 6.
- 12. In the study, many factors were investigated. But it should be pointed out that the results analysis is quite simple. More in-depth analysis is required.
Answer: Additional research was carried out (suggested by the Reviewer) allowed by the time allocated for reviews. Other more complicated studies, related to extended analysis, would require more time and could be a separate topic for another article.
- Comparison of the current developed method with other extraction methods should be included.
Answer: Precipitation was chosen as the reference method because it is currently the most frequently chosen method for extracting antihypertensive drugs from a biological sample (lines 113-114)—supplemented literature in the text.
- The conclusion part should be revised, author make a clear conclusion highlighting the novel points precisely.
Answer: It has been changed.
- There are many typo and grammatical errors in the manuscript. The authors should revise it carefully. English polish is also necessary.
Answer: Error correction has been done.
Round 2
Reviewer 3 Report
Revised manuscript was well improved to accept.